# Retail Food Environment around Schools in Barcelona by Neighborhood Socioeconomic Status: Implications for Local Food Policy

**DOI:** 10.3390/ijerph20010649

**Published:** 2022-12-30

**Authors:** Catalina Londoño-Cañola, Gemma Serral, Julia Díez, Alba Martínez-García, Manuel Franco, Lucía Artazcoz, Carlos Ariza

**Affiliations:** 1Agència de Salut Pública de Barcelona, Public Health Agency, 08023 Barcelona, Spain; 2Departament de Ciències Experimentals i de la Salut (DCEXS), Universitat Pompeu Fabra, 08003 Barcelona, Spain; 3Ciber de Epidemiología y Salud Pública (CIBERESP), 28029 Madrid, Spain; 4Institut d’Investigació Biomédica Sant Pau (IIB Sant Pau), 08041 Barcelona, Spain; 5Public Health and Epidemiology Research Group, School of Medicine, Universidad de Alcalá, Alcalá de Henares, 28801 Madrid, Spain; 6Department of Community Nursing, Preventive Medicine and Public Health and History of Science, University of Alicante, 03690 Alicante, Spain; 7Department of Epidemiology, Johns Hopkins Bloomberg School of Public Health, Baltimore, MD 21205, USA

**Keywords:** food environment, food access, nutrition environment measures surveys, health inequalities, socioeconomic status, geographic information system, schools, children

## Abstract

Childhood obesity is a relevant public health problem. The school food environment has been identified as an important factor for promoting healthy eating behaviors. This study assessed the availability of and proximity to unhealthy food stores around schools (*n* = 22) in the city of Barcelona and its association with neighborhood socioeconomic status (NSES). We conducted this cross-sectional study between 2019 and 2020. First, we identified all food retailers (*n* = 153) within a 400-m buffer around each school and identified those selling unhealthy food products. Then, we used Poison regression models to measure the association between NSES and the healthy food availability index (HFAI), adjusting for population density and distance. A total of 95% of the food establishments studied were classified as unhealthy (*n* = 146). In all, 90% of schools that had, at least, two unhealthy retailers in their proximity. There were significant differences in the mean distance to unhealthy establishments according to neighborhood SES and population density (*p* < 0.05). We found a positive association between schools located in higher SES neighborhoods and a higher availability and affordability of healthy food products (IIR = 1.67, 95% CI = 1.45–1.91 *p* = 0.000). We found strong social inequalities in the supply of healthy foods in Barcelona. Local food policy interventions addressing retail food environment around schools should consider socioeconomic inequalities.

## 1. Introduction

Currently, childhood obesity is one of the main public health problems. In the last four decades, the prevalence of childhood obesity has multiplied by 10 worldwide [1,2]. In Spain, 1 in 10 children presents with obesity (10.3% in 2–17 years of age) [3]. Furthermore, according to data from the World Health Organization (WHO) European Childhood Obesity Surveillance report, Spain is among the countries with the highest prevalence of overweight and obesity [4].

Children’s eating behaviors are shaped by different physical, sociocultural, economic, and political influences, such as the obesogenic environment—defined by Swinburn et al. as the sum of influences from the environment and the opportunities, or living conditions, which promote obesity in individuals or populations [5]. One key element of an obesogenic environment is an unhealthy retail food environment, which is characterized by the availability, accessibility, and promotion of unhealthy food and beverage products in the surrounding food stores [6]. Thus, the retail food environment around schools and play spaces has been considered as a key leverage point for childhood obesity prevention [7].

The school food environment has been described as an important scenario in the development of students’ eating habits [8], whereby changes in the school food environment can improve children’s dietary behavior, body mass index (BMI), and influence the reduction in childhood obesity [9]. Recent systematic reviews have compiled the evidence on the relationship between the school food environment and the nutritional status of children and adolescents, highlighting the importance of the school food environment and the quality of food available in or near the school [9,10,11]. Most of these studies are conducted in the Anglo-Saxon context, with little evidence for the European context, especially in the southern region.

Several studies describe the relationship between the nutritional status of minors and the density of unhealthy food establishments in their environment. For instance, a study in England reported a higher density of fast-food, high-calorie-dense, and other unhealthy food establishments in those neighborhoods with a higher prevalence of overweight or obese children [12]. Vandevijvere (2016) demonstrated that, in New Zealand, six out of ten city schools had an unhealthy food establishment within a walking distance [13]. Moreover, the typology of food establishments was related to the BMI of schoolchildren aged 6–17 years. The greater the availability of supermarkets close to the school area, the lower the BMI among adolescents, whereas the greater the density of convenience stores, the higher the BMI [14].

The geographical environment has been shown to influence the number of retail food establishments [15]. In Spain, a study conducted in the region of Madrid showed an association between the distribution of food establishments in the proximity of schools, and the neighborhood socioeconomic status (NSES), this last factor being determinant over the prevalence of unhealthy food establishments in school surroundings. Schools located in neighborhoods with a lower socioeconomic status (SES) counted a higher number of unhealthy food establishments in their proximity [16].

The obesogenic environment has generally been studied and characterized using questionnaires and geographical information systems. Several studies have used different metric tools to assess and identify obesogenic food environments. These have shown significant differences between the classification of the different outlets; however, it is difficult to compare the evidence and scientific characterization of the environment, due to the lack of internationally comparable and validated tools [17,18,19,20]. Several authors also point out that these tools need to be complemented by other food environment assessment methodologies, integrating a more qualitative approach (e.g., food surveys, interviews, or focus groups), which helps to understand the problem more comprehensively [17,18,21]. However, being faced with an obesogenic food environment may be related to health outcomes such as obesity [22,23].

One of the best and recent tools to assess the food environment is the Nutritional Environment Measures Survey in Stores (NEMS-S). Martinez-Garcia et. al. adapted it for the urban Mediterranean context (NEMS-S-MED) in Spain. The survey measures the establishments’ healthfulness through an observational checklist assessing the availability and cost of healthier versus poorer feeding options [24].

In the present study, this validated instrument is replicated to evaluate the nutritional environment in the city of Barcelona.

Therefore, our objective was to assess the availability of and proximity to unhealthy food around schools in the city of Barcelona, and to study their association with neighborhood socioeconomic status.

## 2. Materials and Methods

### 2.1. Study Design

We conducted this cross-sectional study in the city of Barcelona (Catalonia, Spain) between 2019 and 2020. This study is within the framework of the Barcelona Childhood Obesity Prevention Project (POIBIN), which is the second phase of the Barcelona Childhood Obesity Prevention Project (POIBA) [25], an intervention designed and evaluated to prevent overweight and obesity in schoolchildren [26].

#### 2.1.1. School Sampling

The city of Barcelona covers a total area of 101.35 km^2^ and has a population density of 16,149 inhabitants/km^2^. In 2019/2020, the population between 0 and 14 years of age was 204,754 (12.5% of the total population) [27]. Barcelona is divided into 10 districts, 73 neighborhoods, and 1068 census sections [28]. For reference, census sections, such as census tracts, are the smallest geographic units for which population data are released in Spain.

Schools were our spatial unit of analysis. We selected 22 educational centers (11 public and 11 subsidized) out of the 100 included in the POIBIN project targeting children in their fourth year of preschool (4–5 years). In Barcelona, 60% of preschool schools are public and residential location can define the school type. According to law, one of the requirements for accessing public education is proximity of residence to the school. For the most part, the established schedule in preschool schools is from 9:00 to 16:30, therefore it is full time [29]. These were sampled according to their spatial location in the city; the area-level socioeconomic status of the school’s neighborhood, the prevalence of obesity (7%) [30], and the school type (public or subsidized, the latter being privately owned and managed, but with a subsidy from the regional government). This resulted in half the schools located in high-SES neighborhoods and half the schools in low-SES neighborhoods.

#### 2.1.2. Retail Food Environment

Our main outcome was the spatial access to retailer selling unhealthy foods and beverages within a buffer of 400-m from schools, a distance previously used by other studies [31,32]. We examined the spatial access in terms of (1) availability (counts) of stores selling unhealthy foods and (2) proximity based on straight line distances in m per street section from the school to each unhealthy food store.

Furthermore, we calculated the travel time distance in minutes (time needed to travel to a food outlet from the school). We obtained all measures through the QGIS software (Desktop version 3.14 “Pi”).

To locate food stores, we obtained the census of premises and economic activities of the city of Barcelona [33]. This secondary database is managed by the Department of Statistics and Data Dissemination of the Barcelona Municipal Data Office and is available free of charge. For this study, we used the version of January 2018 [33]. A total of 8154 premises with food activities were identified, that is, retail sales of food products classified according to their activity as: (1) supermarkets; (2) convenience food stores; (3) fruit and vegetables stores; (4) butcheries; (5) fishmongers; (6) bakeries; (7) other specialized food stores.

Out of these, we excluded retailers outside the 400-m buffer for each school obtaining a total of 2047 establishments. Then, we calculated the sample size for our on-field observations assuming a 95% confidence level a probability of 50% (maximum uncertainty assumption), a sampling error of 8%, and a 10% loss for establishments that could not be visited. Our final sample was a total of 153 food stores to be visited in the field.

We gathered data on the retail food environment through direct observations. Therefore, we used the NEMS-S-MED [24]. This is an observational instrument, adapted from the original tool developed by Glanz et al. [34], which assesses the availability and affordability of healthy versus less healthy foods for 12 food groups: (1) fresh fruits; (2) vegetables; (3) nuts; (4) non-alcoholic beverages; (5) bread, cereals, and baked goods; (6) milk and dairy products; (7) eggs; (8) oil and butter; (9) rice; (10) legumes; (11) meat and meat products; and (12) fish and fish products.

We collected all data in the Open Data Kit, a freely accessible web application that allows gathering on-site information with portable devices (Android smartphones) with a wireless network connection [35].

Out of these measures, a Healthy Food Availability Index (HFAI) (Appendix A) score was calculated for each store, following the procedures developed by Glanz et al. and Franco et al. [34,36]. This index assigns points based on the presence of all categories in this shopping basket and additional points for healthier versions of those foods. The scoring system of the HFAI ranges from 0 to 49 points. Specifically, food stores scoring ≤ 36 was defined as unhealthy.

Based on the overall score of each food store, previous studies carried out in the Mediterranean context [16,37], and the methodology used by Alyssa Ghirardelli et al. [38], we classified “unhealthy food stores” as those offering products of high caloric density and low nutritional value such as sugar-sweetened beverages, industrial bakery products, chocolates, candies, or sweet or salty snacks such as potato chips, among others.

#### 2.1.3. Area-Level Covariates

In line with previous studies [26,39], we used the Disposable Family Income Index (DFII) to assess NSES. The DFII combines five variables weighted by different criteria: educational level (measured by the number of university graduates); employment situation (as a ratio of unemployed to working-age population); number of cars with relation to the population; power of new cars purchased by residents; and second-hand residential market prices [40]. The index acts as an indicator of the relative income of residents in the different neighborhoods and provides a snapshot of social inequalities referenced to a city average value centered on 100.

For this study, we classified schools as high-NSES if the DFII was ≥85 and as low-SES if DFII < 85. We used the DFII as a categorical variable based on quintiles of the Barcelona Disposable Family Income distribution [40].

We also obtained data on population density at the neighborhood level, defined as the number of residents per square kilometer within each neighborhood. We used QGIS (Desktop version 3.14 “Pi”) to calculate the area within each neighborhood in square kilometers (sq. Km.). Population data were obtained from the Catalonian Statistics Institute (Idescat) for 2016 [27].

### 2.2. Statistical Analysis

We conducted a descriptive analysis to study the distribution of (1) the density around schools (within buffers of 400 m) of unhealthy food and (2) the distances from the schools to the nearest unhealthy food outlet. Therefore, we used the double-sided Kruskal–Wallis tests (CI 95 to compare medians of both measures by different school characteristics (school type, quintiles of NSES, and tertiles of population density). We completed this for all types of unhealthy food outlets.

To control for factors related to the urban environment, and in line with previous studies that have also explored the relationship between spatial access to food premises and the food environment [16,41], we included population density in our models by dividing the number of residents by the neighborhood area (103 residents/km^2^).

The association between HFAI and neighborhood SES was examined using the Poisson regression model. The main independent variable, neighborhood SES, was treated as a categorical variable based on tertiles (low, middle, high). Our model was adjusted for population density and distance. Subsequently, the availability and affordability of healthy food in each grocery store was illustrated for each SES category (reference category: middle level neighborhood SES).

All analyses were performed with STATA/SE 15 software (Stata Corp, College Station, TX, USA).

## 3. Results

Table 1 shows the frequency of establishments according to store type for Barcelona as a whole, for the sample studied, and for the establishments classified as “unhealthy”. Out of the 153 food establishments located around the 22 schools targeted in this study, 44% of the establishments were located around public schools and the rest around subsidized schools. The largest number of establishments observed were convenience stores (23.5%), followed by butcher or egg stores (20.9%) in line with the frequencies found for the city of Barcelona as a whole. A total of 1% of the establishments corresponded to other specialized food stores (a wine shop and a bar).

According to the HFAI, 95% of the food establishments studied were classified as unhealthy (*n* = 146). Only seven food establishments were classified as healthy according to their typology: one butcher’s shop and six supermarkets.

### 3.1. Counts, HFAI, and Distance of Unhealthy Food Stores

We found that 90% of the schools have at least two unhealthy establishments within a distance inferior to 400 m. Table 2 shows the availability, HFAI score by school type, NSES, population density, and distance from schools to unhealthy establishments (*n* = 146). The number of unhealthy food outlets around schools in low and middle-lower SES areas was higher than in high and middle-high SES areas (64 and 27 unhealthy establishments, respectively). We found significant differences in the mean distance to unhealthy establishments according to NSES and population density (*p* < 0.05). Schools located in neighborhoods with lower SES were in closer proximity to unhealthy establishments than schools located in neighborhoods with higher SES (median = 248 m and 406 m, respectively). As for the HFAI, we found no significant differences by type of school, neighborhood SES, and population density (*p* > 0.05).

Figure 1a shows the spatial distribution of the schools, the total number of food establishments studied (healthy and unhealthy) in the area of influence (400 m) and the neighborhoods’ SES (dark blue tones correspond to areas with higher SES, while the white color is for areas with lower SES). Schools located in central areas of the city have a higher density of establishments, compared to those located in peripheral areas. Figure 1b shows the proximity of unhealthy establishments to the schools according to the neighborhoods’ SES at three distance scales calculated in meters per street stretch. Most establishments selling unhealthy food products were located less than 318 m from the school.

### 3.2. Association between the Healthy Food Availability Index (HFAI) and the NSES

Figure 2 shows the associations between NSES of schools at the neighborhood level and the HFAI scores of each food establishment, measured according to the NEMS-S-MED questionnaire, adjusted for population density and distance. This figure illustrates the availability and affordability of healthy food products in each grocery store by neighborhood SES using the average SES category as the reference. We observed a positive association between higher HFAI scores, and schools located in the least deprived neighborhoods relative to schools located in the reference neighborhood SES category. The availability and affordability of healthy foods were higher in schools located in neighborhoods with higher SES (IIR = 1.67; 95% CI = 1.45–1.91 *p* = 0.000). Conversely, those establishments with lower availability and affordability of healthy food products were more likely to be located near schools in more deprived neighborhoods; however, this trend was not significant in our model (IIR = 1.08; 95% CI = 0.96–1.23 *p* = 0.183).

## 4. Discussion

The main results of the study suggest that in the city of Barcelona 95% of premises within 400 m of a school are unhealthy. Moreover, the HFAI scores obtained from the NEMS-S-MED tool were related to the NSES. Schools located in more deprived neighborhoods had a lower availability of establishments selling healthy food products in their surroundings. Our spatial analysis detailed the availability of grocery stores in the proximity of schools, showing an increased concentration of establishments in the city center, where the population density is higher, and a lower average distance between unhealthy establishments. On the other hand, the majority of schools located in low SES neighborhoods had a lower population density and a greater average distance between healthy premises.

The results of this study are consistent with those reported in both national and international studies [13,16,42]. Díez J et al. found that schools in more disadvantaged areas had an increased exposure to unhealthy retail outlets, as this type of store was located in closer proximity to educational areas. This study was conducted in the city of Madrid, an urban environment that shares similarities to the present study [16]. These findings contribute to the notion that there is a relationship between the availability (count) of stores selling unhealthy food products in the immediate food environment of schools and neighborhood SES. However, this association was not significant in our study. This outcome might be explained by the difference in the study sample, being that the study performed in Madrid had a comparatively much larger sample than the present analyses and was also not conditioned by the school-based childhood obesity prevention project (POIBIN). In contrast, our study identified a low availability of healthy premises (4.5%) indicating a predominant obesogenic food environment in proximity to Barcelona’s school surroundings, which is related to the district’s socioeconomic status.

A study conducted in New Zealand in 2015 also observed a significant obesogenic environment near schools which had become areas with high access to unhealthy food products. This study also reported social inequalities in the access to unhealthy food products [13]. Pinheiro AC et al. in 2022 pointed out the relationship between the availability of establishments and socioeconomic determinants of poverty; in particular, the one that marks the situation of unhealthy premises in the Chilean context [42].

Another important finding of this study is that neighborhoods with lower SES and areas with lower population density had a greater average distance to healthy premises. Therefore, the neighborhood SES influences the geospatial location of unhealthy establishments. Likewise, population density may characterize the dispersion of unhealthy premises. In a similar study carried out in Barcelona for alcoholic beverage outlets, they concluded that the urban environment is characterized by elements of promotion, availability, and consumption in the unequal distribution of the territory [43]. Food establishments also use these promotional strategies to increase consumer demand, especially in the case of schoolchildren and their families [38,44]. Regulating these elements can contribute to improving health-related aspects and access to healthy food products [43,45].

According to the results of our study, the availability of unhealthy food products in the proximity of schools is high and the availability of healthy food products in these retail outlets is mainly low. Some of the research has highlighted that the reduction in the availability of healthy foods in the school environment would help to reduce the prevalence of overweight and obesity in adolescents [10]. In addition, regulations in the school food environment that limit the availability and accessibility of calorie-dense and nutrient-poor foods should be adopted [11]. Individuals who frequent lower-quality food environments may have poorer diets [46] and, in consequence, be more susceptible to diet-related diseases. Studies have reported associations between the proximity to food retail establishments and BMI, with areas with higher calorie-dense food availability having higher rates of obesity [11,19]. This suggests that modifying the typology of food retail establishments could have had an impact on Mediterranean dietary patterns. In recent decades, due to demographic, economic, and social characteristics, traditional (or retail) shops selling fresh and healthy food products have been disappearing [37,47,48]. This study supports evidence from previous observations on the distance set to assess the food environment [16,31,49], with 400 m being the equivalent distance of a 5-min walk [50]. We assessed this distance and found that most of the schools in our study (90%) have at least two unhealthy shops in their proximity. Some studies have used a radius of observation of up to 800 m to assess the food environment, however, these have been conducted in the Anglo-Saxon context, where distances to schools are greater and traveling is usually by different means of road transport, a context far from what occurs in an urban environment in a Mediterranean country [19].

Our study has several limitations. First, we selected educational centers based on a convenience sampling approach to encompass different casuistry in terms of NSES, school ownership, and placement. This constrains the generalizability of the findings to the whole city of Barcelona and influences the selection of the establishments linked to these schools. Secondly, we cannot rule out an observer bias. Different observers participated in the data collection, which may have also influenced the results obtained. However, we trained the observers through specific and protocolized training for data collection. All the information was systematized, and each observer oversaw collecting data for the same school from the beginning to the end, thus minimizing this bias. Finally, the ecological fallacy was associated with the data in this study, where our main unit of analysis is schools and results cannot be inferred at the individual level.

Despite these limitations, our study also presents the following strengths: first, few studies have addressed food environments around schools in the Mediterranean context and, specifically, in the city of Barcelona. Secondly, we employed a previously validated tool, allowing for the comparison of study outcomes between Madrid and Barcelona. Finally, the fieldwork carried out has been rigorous and intense in its scope of action, making it possible to focus on and highlight those points with possible actions based on the design, implementation, and evaluation of public policies, aiming to increase the availability and proximity of healthy foods in the population [51].

Future research should combine different methodologies for data collection, whether individual, contextual, or specific geospatial tools, to better understand the development of the problem in the proximity of schools. In addition, the research should facilitate the participation of other stakeholders such as municipal government, schools, health centers, and local communities. All of these are necessary for the development of food and nutrition policies, their implementation, and evaluation.

To reduce childhood obesity and related inequalities, the Childhood Obesity Task Force in London has devised food strategies aimed at reducing children’s exposure to junk food and restricting advertising of unhealthy foods. In addition, the London Food Strategy proposes concrete measures to act within 400 m of schools by limiting take-away food outlets [52]. In light of the data from this study, it is important that Barcelona considers a similar policy around its schools.

## 5. Conclusions

A total of 95% of the food establishments in the city of Barcelona located in the proximity of schools were classified as unhealthy, and this phenomenon was more frequent in deprived neighborhoods. We found a factor of inequality in the supply of healthy food products: the availability and affordability of healthy food were higher in schools located in neighborhoods with a more favorable socioeconomic status and, on the contrary, it was lower in schools located in neighborhoods with disadvantaged SES.

Local food-policy interventions should consider socioeconomic inequalities in the proximity and availability of food facilities near schools.

## Figures and Tables

**Figure 1 ijerph-20-00649-f001:**
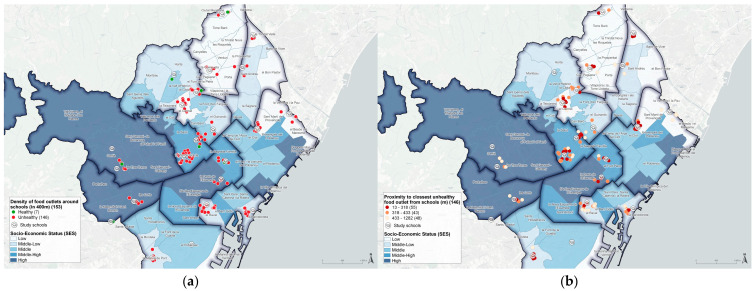
Schools in the city of Barcelona (2019) by neighborhood socioeconomic status (SES): (**a**) shows the availability of retail outlets around schools (within a 400-m radius); (**b**) shows the proximity of unhealthy retail outlets around schools (within a 400-m radius).

**Figure 2 ijerph-20-00649-f002:**
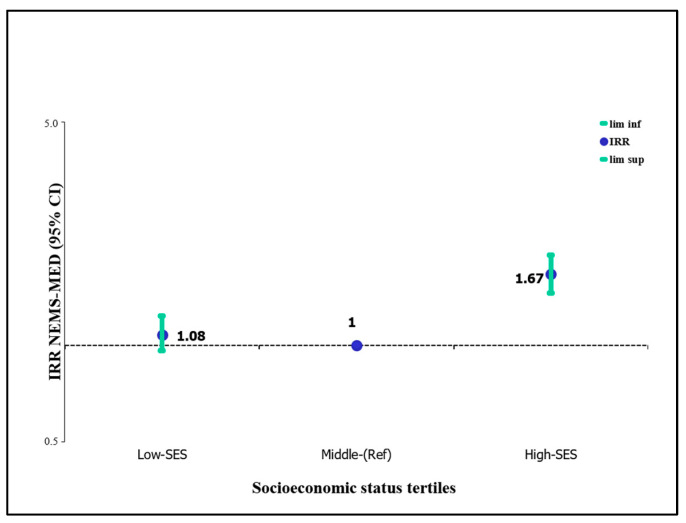
Association between the Healthy Food Availability Index (HFAI) of each food establishment and the socio-economic status of schools at the neighborhood level using Poisson regression. Model adjusted for population density and distance.

**Table 1 ijerph-20-00649-t001:** Food retailers by store type found within 400 m of schools in Barcelona as a whole, overall and for the sample of establishments classified as unhealthy (Barcelona, 2019).

	Total Barcelona		Total Study		Unhealthy Food Stores	
Store Type	*n*	%	*n*	%	*n*	%
Supermarkets	236	11.53	18	11.76	12	8.22
Convenience food stores	480	23.45	36	23.53	36	24.66
Fruit and vegetables stores	347	16.95	26	16.99	26	17.81
Butcheries	439	21.45	32	20.92	31	21.23
Fishmongers	207	10.11	16	10.46	16	10.96
Bakeries	309	15.10	23	15.03	23	15.75
Other specialized food stores	29	1.42	2	1.31	2	1.37
Total	2047	100	153	100	146	100

**Table 2 ijerph-20-00649-t002:** Counts, Healthy Food Availability Index (HFAI), and distance (m) to unhealthy retailers (*n* = 146) across schools citywide (Barcelona, 2019).

Characteristics	Availability	Healthy Food Availability Index (HFAI)		Distance ^3^	
	*n* (%)	Median (IQR) ^1^	*p*-Value ^2^	Median (IQR) ^1^	*p*-Value ^2^
School type			0.1035		0.0558
Public	64 (43.84)	7 (2–13)		318 (170–453)	
Subsidized	82 (56.16)	4 (2–9)		390 (294–474)	
NSES			0.2974		0.0001
Low-SES	34 (23.29)	7 (3–15)		248 (166–453)	
Middle-low	34 (23.29)	3 (2–7)		469 (411–525)	
Middle	51 (34.93)	7 (2–12)		369 (292–426)	
Middle-high	9 (6.16)	5 (3–8)		177 (165–264)	
High-SES	18 (12.33)	7 (2–17)		406 (150–482)	
Population density (103 residents/km^2^)			0.1651		0.0164
Low	44 (30.10)	7 (2–15)		363 (238–466)	
Medium	55 (37.70)	6 (2–11)		338 (198–415)	
High	47 (32.20)	4 (2–8)		443 (264–526)	

^1^ IQR = interquartile range; ^2^
*p*-values correspond to Kruskal–Wallis test; ^3^ Distance is calculated in meters by street section. NSES: neighborhood socioeconomic status.

## Data Availability

The data presented in this study are available on request from the corresponding author.

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
