# Peer review of "Retail Food Environment around Schools in Barcelona by Neighborhood Socioeconomic Status: Implications for Local Food Policy"

_ijerph, 2022, doi:10.3390/ijerph20010649_

Round 1
Reviewer 1 Report
The study is an interesting starting point for studying the food environment around schools. As the study itself points out, these are initial analyzes and further studies should be conducted in the area to better understand the food context and how it influences the health of the population.
Some points, however, can be improved. First, it is important to contextualize the reader who is not familiar with the education system in Barcelona. What percentage of children study in public schools in barcelona? Does the place of residence define the school that the children will study? Children between 4 and 5 years old study full time? In general, how far are homes from schools?
Some additional analyzes could be conducted. As you have the adress of schools and stores geocoded, it would be interesting to assess how schools influence the surrounding commerce, even without considering whether these stores are healthy or unhealthy . There are specific analyzes that can demonstrate whether schools act as a magnet to attract food businesses.
Other specific points to be improved:
Page 3, line 112: Include characteristics of selected schools. How many were public, how many were located in low and high income neighborhoods, and prevalence of obesity.
Table 1: The formatting of the table makes it difficult to visualize the information. it is important that the numbers are aligned with the types of stores.
Figure 1: Improve resolution. Even applying the PDF the resolution is very bad.
Reviewer 2 Report
Thank you for the opportunity to review this work. The authors have assessed the availability and proximity to unhealthy food around schools in the city of Barcelona and to study their association with neighborhood socioeconomic status (NSES). The findings show that 95% of the premises located within 400 m of a school were unhealthy. The authors reported that most of the schools located in a low SES neighborhood had a lower populations density and greater distance between healthy premises. Overall, it is a well written paper and the authors have examined a very important issue. Childhood obesity is becoming a global problem of cause and concern and it is very interesting that the authors chose to examine the availability of unhealthy food premises near schools. It was interesting to observe the difference in food availability based on SES. Great use of maps in Figure 1.
Minor comments:
· Have similar studies been done in populations living in lower income countries? It would be interesting to see how the findings compare if such studies are there because in several of these countries’ childhood obesity is slowly beginning to emerge as a problem.
· Why was a distance of 400 m considered between schools and food stores?
· It would be interesting to see if additional such studies could lead to a policy change in promoting healthy eating among children and adolsecents and bring about a change in food choices.
· Please double check the numbers in the tables which should have a decimal. These numbers seem to have a comma. This change needs to be made throughout.
Reviewer 3 Report
This manuscript looks at the impact on children's healthy eating by evaluating the eating environment around schools in communities of different socio-economic status. The research is quite interesting.
1.The government building plans around schools by neighborhood socioeconomic status is also particularly important for the layout of retail outlets around schools. It is recommended that it be taken into account.
2.For the method NEMS-S-MED, the impact of the retailer's marketing strategy or brand effect on the assessment score is not considered. It is recommended that it be taken into account.
3.Can the conclusions be applied elsewhere because of the specificity of the sample population? It is recommended that the authors give further clarification.
